# The Multifunctional Role of EMP3 in the Regulation of Membrane Receptors Associated with IDH-Wild-Type Glioblastoma

**DOI:** 10.3390/ijms22105261

**Published:** 2021-05-17

**Authors:** Antoni Andreu Martija, Stefan Pusch

**Affiliations:** 1Clinical Cooperation Unit (CCU) Neuropathology, German Cancer Consortium (DKTK), German Cancer Research Center (DKFZ), 69120 Heidelberg, Germany; a.martija@dkfz-heidelberg.de; 2Department of Neuropathology, Heidelberg University Medical Center, 69120 Heidelberg, Germany; 3Faculty of Biosciences, Heidelberg University, 69120 Heidelberg, Germany

**Keywords:** EMP3, glioblastoma, membrane receptors

## Abstract

Epithelial membrane protein 3 (EMP3) is a tetraspan membrane protein overexpressed in isocitrate dehydrogenase-wild-type (IDH-wt) glioblastoma (GBM). Several studies reported high EMP3 levels as a poor prognostic factor in GBM patients. Experimental findings based on glioma and non-glioma models have demonstrated the role of EMP3 in the regulation of several membrane proteins known to drive IDH-wt GBM. In this review, we summarize what is currently known about EMP3 biology. We discuss the regulatory effects that EMP3 exerts on a variety of oncogenic receptors and discuss how these mechanisms may relate to IDH-wt GBM. Lastly, we enumerate the open questions towards EMP3 function in IDH-wt GBM.

## 1. Introduction

The diagnosis of glioblastoma (GBM) is applied to highly aggressive primary central nervous system (CNS) tumors. Histologically, GBM is characterized by diffusely infiltrating tumor cells with nuclear atypia, high mitotic index, microvascular proliferation, and/or necrosis [1,2]. It is known for its dismal prognosis and poor overall response to the standard therapy of surgical resection, radiation, and chemotherapy [3,4]. In the last years, several tumor types are subsumed under this term requiring further distinction. A main division is drawn between isocitrate dehydrogenase (IDH)-mutant GBM, usually evolving from IDH-mutant astrocytoma, and IDH-wild-type (IDH-wt) GBM. This review addresses IDH-wt GBM.

IDH-wt GBM is defined by its heterogeneous molecular landscape [5,6,7,8,9,10,11,12,13,14]. Two landmark studies have subclassified IDH-wt GBM based on genetic, epigenetic, and transcriptomic features. Utilizing samples from The Cancer Genome Atlas (TCGA) network, Verhaak et al. described four different subtypes of IDH-wt GBM: classical (CL), mesenchymal (MES), proneural (PN), and neural (NE) [5]. The NE subtype has since been discarded, as later findings determined this group to be non-tumor specific [12]. Using another approach, Sturm et al. classified IDH-wt GBM into five epigenetic groups on the basis of DNA methylation patterns [6]. These groups have been further modified and expanded by Capper et al. [7] into seven methylation subclasses, each defined by a distinct set of molecular alterations [3,6,7] (Figure 1). Three of the most common methylation subclasses among adult patients—RTK I, RTK II, MES—are enriched for the PN, CL, and MES Verhaak expression profiles, respectively [6]. These tumor entities typically exhibit chromosome 7 gain (with or without epidermal growth factor receptor or *EGFR* amplification), as well as loss of chromosomes 9q21 and 10 [3,6,7]. Despite sharing common chromosomal alterations, these groups can be further differentiated by subclass-specific changes: platelet-derived growth factor receptor α (*PDGFRA*) amplification in RTK I, gain of chromosomes 19 and 20 in RTK II, and neurofibromin 1 (*NF1*) mutations in MES tumors [3,6,7].

Similar to the first three subclasses, RTK III tumors frequently exhibit chromosome 10 loss along with *EGFR* amplification; however, this diagnosis is more common among pediatric patients compared to RTK I, RTK II, and MES tumors [7]. The MYCN and midline (MID) subclasses, which are also mostly pediatric, harbor distinct alterations in *MYCN* and fibroblast growth factor 1 receptor (*FGFR1*), respectively [7]. In addition, the MID subclass frequently exhibits *PDGFRA* amplification and loss of the cyclin-dependent kinase inhibitor 2A/B (*CDKN2A/B*) [7]. IDH-wt GBM also currently includes gliomas with characteristic mutations in histone 3 of the G34K/G34V type. These tumors will be allotted a distinct tumor type in the upcoming WHO 2021 (unpublished, personal communication AvD) and will not be addressed here further. Methylation-based tumor diagnosis has recently attained much attention [7] and is currently introduced in diagnostic routine worldwide. Nevertheless, we are using expression-based subclassification for our subsequent considerations as it is most frequently used in the summarized publications.

Most of the molecular alterations in GBM are mutations in genes coding for components of receptor tyrosine kinase (RTK) signaling pathways [15] and have been reviewed in great detail by Weller et al. [3] and Pearson et al. [4]. Apart from genetic mutations, transcriptional alterations have also been identified to be putative GBM contributors [5,6,9]. These include the transcript coding for epithelial membrane protein 3 (EMP3), which is frequently overexpressed in IDH-wt GBM based on several studies [16,17,18,19]. The reproducible occurrence of EMP3 overexpression, along with several analyses linking it to poor clinical outcomes [16,17,20,21,22], hint towards an active role of EMP3 in these tumors. In this review, we summarize current literature about EMP3 and its involvement in IDH-wt GBM. We highlight its regulatory effects on several oncogenic membrane proteins and discuss how these mechanisms may contribute to the development of this dreaded disease.

## 2. EMP3 Biology

### 2.1. EMP3 Protein Structure and Localization

EMP3 is a 163-amino acid tetraspan membrane protein (~18 kDa) encoded by the *EMP3* gene in the human chromosome 19q13.3 [23,24,25]. It belongs to the peripheral myelin protein 22-kDa (PMP22) gene family based on the sequence and structural homology [24,26]. This group of proteins, which also includes epithelial membrane proteins EMP1 and EMP2 [23,24], has been implicated in the regulation of cellular survival and cancer, as well as tissue-specific biological processes like chondrocyte differentiation, angiogenesis, and peripheral nerve myelination [24]. At the subcellular level, PMP22 family members have been linked to the regulation of membrane organization, cell-to-cell interactions, and signal transduction processes [26,27]. 

The major part of EMP3 consists of four hydrophobic membrane-spanning domains spanning 21 amino acids each [25,27] (Figure 2). There are two extracellular domains (ECD) of 41 and 18 amino acids each [25,27]. The three intracellular domains consist of short cytoplasmic N-and C-terminal tails and a loop of 13 amino acids [25,27]. Its hydrophobic regions, particularly the 2nd and 4th membrane-spanning domains, are highly conserved among the PMP22 protein family [24,26]. In addition, EMP3′s domains are assumed to be subject to several post-translational modifications (PTMs). For example, EMP3′s first ECD is predicted to include two *N*-glycosylated residues at asparagine 47 (N^47^) and 56 (N^56^) [24,25,27]. Between these two residues, only N^47^ has been experimentally confirmed to be necessary and sufficient for glycosylation, indicating that the attachment of *N*-linked glycans solely occurs at this residue or that *N*^56^-glycosylation requires the presence of asparagine at position 47 [25]. The glycan modifications attached to EMP3 are diverse, because several bands migrating between 20 and 30 kDa, all amenable to enzymatic deglycosylation, have been detected by immunoblotting [25]. While the functional relevance of *N*-linked glycosylation on EMP3′s function remains to be determined, current knowledge on glycoproteins suggests that this PTM could be important for protein homeostasis, cell adhesion, or cell-to-cell communication [28,29]. In addition to being glycosylated, EMP3 is predicted to harbor a protein kinase phosphorylation site at threonine 90 within its intracellular loop [24]. However, the existence and functional relevance of this predicted PTM are yet to be confirmed.

In terms of its subcellular localization, EMP3 is predicted to reside in the plasma membrane and within cytoplasmic vesicles [30]. This has been substantiated by immunostaining experiments, which showed both membrane and cytoplasmic pools of EMP3 [17,25]. However, until now, precise mapping of its cytoplasmic or vesicular localization remains elusive. Christians et al. reported that intracellular EMP3 only partially co-localizes with the Golgi marker RCAS1 and does not localize at all with the lysosomal marker LAMP1 and the endoplasmic reticulum marker calnexin [25]. The same study, however, reported EMP3′s interaction with several proteins that localize in vesicles or endosomes [25], indicating that intracellular EMP3 may instead localize within these membrane-bound trafficking compartments. 

### 2.2. Cell Type- and Tissue-Specific Distribution of EMP3

EMP3 is ubiquitously expressed and is notably enhanced in certain tissue types [26]. Expression data from the Human Protein Atlas (http://www.proteinatlas.org, accessed on 20 April 2021) indicate that EMP3 is enhanced in blood, with expression levels being particularly high in granulocytes, monocytes, and lymphocytes [31]. Its enrichment in monocytes is further confirmed by single cell RNA sequencing (scRNA-seq), which showed high physiological expression of EMP3 in macrophages and specialized macrophages like Kupffer cells in the liver and Hofbauer cells in the placenta [31]. In erythrocytes and erythroid progenitors, the EMP3 protein corresponds to the MAM surface antigen [27]. Apart from being expressed in blood cells, EMP3 levels are also documented to be high in several epithelial and mesenchymal cell types including melanocytes, keratinocytes, and fibroblasts [31].

Compared to blood and other tissues, EMP3 expression in the adult brain appears to be relatively low and has little regional specificity [26,31]. This is further confirmed by a recently published single-nucleus RNA sequencing (snRNA-seq) dataset of the adult human brain, which shows very low EMP3 expression across most cell types in the cortex, with the exception of endothelial cells, astrocytes, and microglia [32]. On the other hand, scRNA-seq of developing human brain organoids have revealed high EMP3 levels in actively dividing progenitor cells, radial glia, and mesenchymal cells [32]. This expression pattern suggests that proliferative capacity and mesenchymal cell identity are tightly linked to EMP3 expression during neurodevelopment.

## 3. EMP3 in IDH-wt GBM

### 3.1. EMP3 Expression in IDH-wt GBM

EMP3 is among the set of non-catalytic proteins that are thought to play a critical role in IDH-wt GBM. Analysis of TCGA data showed that EMP3 expression is higher in IDH-wt GBM compared to normal brain tissue and IDH-mutant gliomas with the glioma CpG island methylator phenotype (G-CIMP) [16,17]. Likewise, Ernst et al. could demonstrate high EMP3 levels in GBM spheroid cultures [19]. Scrideli et al. also found high EMP3 expression in GBM which likely were IDH-wt [18]. In contrast, a study from the pre-IDH era reported on low expression of EMP3 in secondary GBM likely corresponding to IDH-mutant GBM, in low-grade gliomas (LGGs), and in non-neoplastic tissue [33]. The low expression of EMP3 in LGGs corresponds to EMP3 promoter hypermethylation, which is not seen in GBM [33]. This suggests epigenetic regulation of EMP3 in glial brain tumors. Histologically, cells with high EMP3 expression were found to be especially frequent within the tumor core [34].

Among GBM Verhaak subtypes, EMP3 expression is highest in MES GBM followed by CL GBM, and is lowest in the PN subtype [5,16,17]. Recent scRNA-seq of GBM tumors also confirm high expression of EMP3 in mesenchymal-like and astrocytic-like cells roughly matching to the MES and CL Verhaak subtypes, respectively [8] (Figure 3).

### 3.2. Prognostic Value of EMP3 in IDH-wt GBM

Several studies have shown that high EMP3 transcript levels correlate with advanced disease and poor clinical outcome. Two independent analyses of TCGA-GBM data have shown that high EMP3 expression in GBM is associated with shorter overall survival [16,17]. In addition, EMP3 was established to be part of a 4-gene signature that can predict the survival of patients with GBM [20]. In this scoring system, high EMP3 expression was found to correlate with a higher risk score and poorer overall survival (OS). Similarly, EMP3 was identified to be part of a 60-gene signature that was differentially expressed between long-term (i.e., survival >4 years) and short-term (i.e., <1 year) survivors in the Memorial Sloan Kettering Cancer Center (MSKCC) and TCGA patient cohorts [21]. In this study, there was a 33% decrease in EMP3 expression in MSKCC and TCGA long-term survivors (LTS) compared to TCGA patients with a survival of less than 1 year. The majority of the genes that were downregulated in LTS were associated with the MES subtype [21], suggesting that EMP3 is part of a mesenchymal transcriptional program that imparts a poorer prognosis. These findings echo an earlier study [22], which performed overlap analysis of four GBM microarray datasets to come up with a robust 9-gene signature that includes EMP3. High expression of EMP3 was correlated with clinically unfavorable metagene expression and overexpression of CD133, nestin, and mesenchymal markers [22]. Collectively, these data point toward a role of EMP3 in affecting clinical outcome in IDH-wt GBM. However, the functional basis of this association remains to be resolved. 

## 4. EMP3-Dependent Regulation of Membrane Receptors

### 4.1. Role of EMP3 in Receptor Tyrosine Kinase Signaling

The best studied function of EMP3 is its regulation of RTK signaling. Activated RTKs catalyze the phosphorylation of tyrosine residues in signaling proteins. Activation is mediated by ligands including growth factors, cytokines, neurotrophic factors, or hormones [4]. RTKs are composed of an extracellular ligand-binding domain, a membrane-spanning hydrophobic region, and a cytoplasmic tail harboring tyrosine kinase domains [4]. RTK-ligand binding typically induces receptor dimerization or oligomerization, followed by subsequent transphosphorylation of the binding partners [4]. This initiates multistep signaling cascades promoting proliferation and differentiation [4,15].

The RTK EGFR is frequently overactivated in IDH-wt GBM. EGFR overactivation may result from *EGFR* gene amplification or the EGFR-vIII mutation, which is defined by an intragenic truncation that leads to constitutive activation of the kinase domain [3,15]. Roughly 40% of all IDH-wt GBM carry EGFR amplification of which approximately half are also carrying the EGFR-vIII mutation. Other RTKs activated in IDH-wt GBM include PDGFRA (amplified in 10–13% cases), the hepatocyte growth factor receptor MET (amplified in 4% of cases), the EGFR family member ErbB2/HER2 (mutated in 8% of cases), the insulin-like growth factor 1 receptor (IGF-1R), and members of the vascular endothelial growth factor receptor (VEGFR) and fibroblast growth factor receptor (FGFR) families [4,15]. Lastly, mutations in downstream RTK signaling effectors and regulators like NF1, PTEN, PI3K, Akt, and Ras have also been documented in GBM [15], with alterations in the first two being particularly enriched in MES GBMs [5,6].

Several studies have linked EMP3 to the regulation of RTKs (Figure 4). Using a human chondrosarcoma cell line, Christians et al. documented a permissive effect of EMP3 on RTK-dependent mitogenic signaling [25]. Small hairpin RNA (shRNA)-mediated knockdown of EMP3 led to decreased levels of phosphorylated EGFR (p-EGFR) and Akt (p-Akt), without greatly affecting total protein EGFR and Akt levels. Furthermore, EMP3 knockdown reduced total and phosphorylated ERK (p-ERK). The inhibition of the ERK and Akt pathways resulting from EMP3 depletion correlated with attenuation of in vitro cellular proliferation and wound healing capacity, as well as increased sensitization to staurosporine- and tumor-necrosis factor related apoptosis inducing ligand (TRAIL)-induced apoptosis [25]. Similarly, using hepatocellular carcinoma (HCC) cells, Hsieh et al. demonstrated significant reductions of p-ERK and p-Akt upon shRNA-mediated knockdown of EMP3 [35]. In their system, EMP3 knockdown did not affect total ERK and Akt levels; instead, Akt inactivation was attributed to a reduction in the levels of p85, the regulatory subunit of PI3K [35]. Conversely, Wang et al. have reported that EMP3 overexpression in TSGH8301 human bladder cancer cells increased p-Akt levels [36]; however, in contrast to the previous study [35], there was a concomitant upregulation of the p110α catalytic subunit of PI3K and no reported effect on the p85 subunit [36]. In line with other findings [25], EMP3 knockdown in HCC cells reduced cell proliferation and attenuated tumor growth in vivo [35], while EMP3 overexpression in TSGH8301 cells promoted cell growth and increased the proportion of Ki-67-positive cells in vitro [36]. In addition, EMP3 knockdown also inhibited HCC motility and invasiveness, an effect that was further attributed to reduction of extracellular matrix (ECM) degraders matrix metalloproteinase 9 (MMP-9) and urokinase plasminogen activator (uPA) [35]. Conversely, EMP3 overexpression promoted cell migration of human bladder cancer cells [36]. The effects on HCC cellular migration were rescued by Akt overexpression, suggesting that EMP3′s invasion-promoting function is primarily mediated by the PI3K/Akt pathway [35]. 

Additional studies have linked EMP3 to the regulation of the EGFR family member ErbB2/HER2. For example, EMP3 overexpression and knockdown in human bladder cancer cells have been shown to culminate into a concomitant increase and reduction in ErbB2/HER2 protein levels, respectively [36]. Interestingly, ErbB2/HER2 knockdown and PI3K pharmacological inhibition also reduced EMP3 protein levels, implying functional cross-talk between EMP3 and the HER2-PI3K/Akt pathway [36]. These results were in line with expression data from primary breast carcinomas, which showed a positive correlation between EMP3 and ErbB2/HER2 expression [37]. 

How exactly does EMP3 regulate RTK-dependent signaling? The protein has no experimentally verified catalytic domains. Furthermore, its cytoplasmic domains, except for its 13-amino acid intracellular loop, are too short to exert a regulatory function. The positive relationship between EMP3 and ErbB2/HER2 protein levels [37] is suggestive of a stabilizing function of EMP3, which is reminiscent of EMP3′s documented effect on a non-RTK membrane receptor [27] and is most likely mediated by EMP3′s extracellular or transmembrane domains. Apart from stabilizing membrane receptors, EMP3 can also regulate mitogenic signaling by modulating the membrane organization and trafficking of RTKs. A yeast two-hybrid (Y2H) screen, for example, has shown that EMP3 interacts with several regulators of EGFR trafficking, most notably FLOT1 and HTATIP2 [25]. FLOT1 is known to facilitate EGFR and ErbB2/HER2 clustering within lipid raft membrane microdomains, thereby promoting downstream RTK signaling [38]. Conversely, HTATIP2 promotes EGFR degradation and attenuates RTK signaling from the cytoplasm by inducing V-ATPase-dependent acidification of EGFR-carrying endosomes [39]. Activation of RTK signaling by EMP3 may thus be facilitated by interaction with FLOT1 or HTATIP2. Further studies will be needed to fully dissect these putative mechanisms and to clarify how other protein partners cooperate with EMP3 to support RTK signaling. More importantly, it will be necessary to understand EMP3-dependent RTK regulation in an IDH-wt GBM setting. 

### 4.2. EMP3-Dependent Regulation of TGF-β Signaling

The transforming growth factor beta (TGF-β) signaling pathway is involved in the regulation of cellular survival, differentiation, migration [40,41]. Similar to RTK signaling, it is initiated upon binding of the TGF-β ligand to the type II receptor serine/threonine kinase TGF-β receptor 2 (TGFBR2) [41]. TGFBR2 then heterodimerizes with and phosphorylates the cytoplasmic domain of its type I counterpart, TGF-β receptor 1 (TGFBR1) [40,41]. With the assistance of adaptor proteins, TGFBR1 phosphorylates the cytoplasmic receptor-regulated Smad proteins (R-Smads) Smad 2 and Smad 3 [40,41]. Activated R-Smads form a heterotrimer with the common-partner Smad 4, and the resulting complex translocates into the nucleus to promote Smad-dependent transcription [40,41]. Alternatively, TGF-β receptors may initiate signaling via Smad-independent mechanisms and activate the Ras/Raf/MEK/ERK, PI3K/Akt, and other growth-promoting pathways [40,41]. TGF-β has been implicated in glioblastoma stem cell (GSC) maintenance, ECM degradation and GBM invasion, vascular remodeling, and immunosuppression [40].

Using a panel of glioblastoma cell lines enriched for the MES GBM marker CD44, Jun et al. have demonstrated EMP3’s ability to modulate TGF-β signaling [17]. Knockdown of EMP3 using shRNAs was shown to attenuate TGF-β signaling in CD44-high GBM cells [17]. In particular, TGF-β treatment of serum-starved EMP3-depleted cells resulted into a lesser increase in Smad 2/3 phosphorylation and transcription of TGF-β target genes compared to EMP3-expressing cells [17]. Furthermore, it was shown that EMP3 interacts with TGFBR2 in response to TGF-β treatment [17]. Collectively, the results are consistent with a model wherein EMP3 interacts with TGFBR2 upon TGF-β stimulation to activate downstream R-Smads (Figure 5). Such an effect is reminiscent of the function fulfilled by the non-catalytic type III receptor TGFBR3, which promotes TGF-β signaling through ligand sequestration and presentation as well as recruitment of cytoplasmic adaptor proteins [41]. Whether EMP3 facilitates TGF-β signal transduction by similar mechanisms remains to be further explored. Regardless, EMP3-dependent activation of TGF-β signaling appears to be pathologically relevant, because EMP3-depleted intracranial GBM xenografts were shown to exhibit reduced tumor growth along with decreased p-Smad 2/3 levels [17].

### 4.3. EMP3 and GBM-Associated Extracellular Matrix Receptors

A critical component of the GBM tumor microenvironment (TME) is the ECM, which consists of a network of secreted molecules that act as a biomechanical scaffold and a biochemical regulator of tumor cell homeostasis. GBM tumor cells are known for their capacity to remodel and move through the ECM, thereby facilitating aggressive tumor growth and infiltration [42]. Integral to these processes are membrane receptors that recognize various ECM components, including integrins and the mesenchymal ECM receptor CD44. 

Integrins are transmembrane receptors which provide a structural link between the cytoskeleton and the non-cellular environment [43,44]. Structurally heterodimeric, these receptors are composed of interacting single-pass α and β subunits [43,44]. Currently, there are 24 documented integrin heterodimers that bind to various ECM components, including collagen, fibronectin, laminin, vitronectin, tenascin, and osteopontin [44]. Binding of integrins to these ligands initiate receptor clustering, followed by activation of focal adhesion kinase (FAK)-Src signaling [43,44]. This signaling cascade allows cells to remodel their cytoskeleton, thereby altering cell shape and architecture in response to migratory cues [43,44].

On the other hand, CD44 is a single-pass, glycosylated, type I transmembrane protein [45] overexpressed in GBM, particularly in the MES subtype [5,8] and in hypoxic and perivascular tumor regions [46]. It is composed of an extracellular globular domain, a stem region that can incorporate multiple variant regions, a membrane-spanning region, and an intracellular domain that harbors various binding motifs for cytoplasmic proteins [45]. CD44′s extracellular region is known to bind to the ECM components hyaluronic acid (HA) and osteopontin [45,47,48]. HA-CD44 ligand binding typically initiates a signaling cascade involving the ezrin-radixin-moesin (ERM) and Rho GTPase family of proteins [45,47]. Like integrin-FAK-Src signaling, CD44 signaling through these cytoplasmic components can lead to cytoskeletal rearrangements that are permissive of cell motility [47]. On the other hand, osteopontin-CD44 binding has been shown to promote cleavage of CD44′s intracellular domain, leading to HIF-2α-dependent hypoxic signaling and maintenance of stem cell-like phenotypes in GBM [48]. Moreover, CD44 has been shown to interact with several RTKs to promote cellular survival [45].

Several studies have linked EMP3 to the regulation of these ECM receptors. Wang et al. have demonstrated that EMP3 overexpression in human bladder cancer cells leads to an upregulation of the expression of a panel of integrins (α1, α2, α3, α5, αV, α6, and β1) at the mRNA level [36]. Furthermore, EMP3 overexpression increased the levels of total FAK and phosphorylated Src (p-Src) and induced the upregulation of the downstream FAK targets ROCK1 and ROCK2 [36]. These biochemical findings correlated well with increased migration of bladder cancer cells upon EMP3 overexpression [36]. Collectively, the findings indicate that EMP3 promotes a transcriptional program that leads to enhanced integrin expression, which subsequently promotes FAK/ROCK-dependent cellular migration (Figure 6). How EMP3 promotes the transcription of integrin-coding genes, and whether EMP3 also regulates integrin function at the protein level remains to be explored in future investigations.

Similarly, Thornton et al. have demonstrated a functional association between EMP3 and CD44 in erythroid cells [27]. Loss of EMP3 appeared to alter the distribution of CD44 in the plasma membrane of dividing erythroid progenitors [27]. In particular, dividing EMP3-negative progenitors exhibited broader CD44 distribution and higher CD44 levels in the cleavage furrow compared to their EMP3-positive counterparts [27]. It is speculated that this results into altered actinomyosin fusion in EMP3-negative cells, leading to enhanced erythroid proliferation and increased reticulocyte yields [27]. Interestingly, CD44 also co-immunoprecipitated with EMP3 and was found to be reduced in EMP3-negative mature erythrocytes [27]. The reduction in CD44 levels was presumed to be due to increased plasma membrane vesiculation as EMP3-negative erythroid progenitors differentiate into erythrocytes [27]. Taken together, the findings indicate that EMP3 regulates erythropoiesis by stabilizing and organizing the distribution of CD44 within the plasma membrane (Figure 7). 

This CD44-stabilizing function of EMP3 may be relevant for IDH-wt GBM as well, because EMP3 and CD44 levels have been positively correlated in paraffin-embedded GBM specimens and in a cellular panel consisting of normal human astrocytes and human GBM lines [17]. Further studies are needed to elucidate the functional relevance of this association in GBM. It will be interesting and perhaps clinically relevant to determine which CD44-dependent signaling pathways require EMP3, and how these pathways impact GBM development.

### 4.4. Potential Involvement of the EMP3-P2RX7 Axis in GBM

P2RX7 is a ligand-gated ion channel that modulates transmembrane ion movement in response to ATP binding [49,50]. Its activity is proposed to be bimodal and is highly dictated by agonist concentration. In the presence of low amounts of ATP, P2RX7 forms a cation-selective channel that simultaneously allows potassium efflux as well as calcium and sodium influx. However, upon sustained ATP stimulation, P2RX7 is converted into a non-selective pore that allows transport of these cations along with larger biomolecules [49]. These opposite modes of activation are proposed to be the basis for the growth-promoting and cytolytic effects of P2RX7, respectively [49].

P2RX7 is increasingly being recognized as contributory to GBM pathogenesis. Immunohistochemistry (IHC) staining of human GBM tissues have shown that P2RX7 expression is generally higher in tumor than in non-tumor samples and increases with higher grade [51,52]. Furthermore, functional studies have shown that P2RX7, if activated by the agonist BzATP or if overexpressed, promotes growth and migration of glioblastoma cells in vitro and in vivo [51,53]. These oncogenic effects are thought to be ERK-dependent, as BzATP treatment led to increased expression and phosphorylation of ERK [51]. Further confirming the existence of a P2RX7 -ERK signaling axis, treatment with an MEK/ERK pathway inhibitor abrogated BzATP-induced cell proliferation and migration [51]. On top of promoting ERK signaling, P2RX7 activation has also been shown to increase the expression of epithelial-to-mesenchymal transition (EMT) markers, thereby promoting GSC invasiveness [54]. Conversely, inhibition of P2RX7 using various P2RX7 antagonists has been shown to inhibit in vitro cell proliferation [55] as well as tumor growth of glioma cells in vivo [52]

To date, the only functional link between EMP3 and P2RX7 is a biochemical description of their interaction. Using co-purification, co-immunoprecipitation, and GST pull-down assays performed on HEK293 cells, Wilson et al. have shown that EMP3 physically and stably interacts with the C-terminus of P2RX7 [56]. In this context, EMP3 is presumed to favor the cytolytic nature of P2RX7, because EMP3 overexpression induced membrane blebbing and increased lactate dehydrogenase (LDH) release, trypan blue uptake, and annexin V binding [56] (Figure 8). This apoptotic-like phenotype was observed to be caspase-dependent, because it was inhibited by the caspase inhibitor z-VAD [56]. Whether this pro-apoptotic effect holds true in the setting of GBM remains to be explored. Given the two proteins’ individual links to GBM development, it is attractive to think that in GBM, EMP3 can shift P2RX7 to its pro-tumorigenic state instead. Further biochemical and functional studies, however, will be certainly required to test this hypothesis. 

### 4.5. Emerging Insights on the Immunological Role of EMP3

The GBM TME includes resident and infiltrating immune cells that exert pro- and anti-tumorigenic effects in response to various regulatory mechanisms. Tumor-associated macrophages (TAMs) constitute the largest proportion of immune cells in the GBM TME [57,58]. These TAMs mostly originate from infiltrating bone marrow-derived monocytes and, to a lesser extent, brain-resident microglia [57,58]. TAMs are highly plastic and may assume a spectrum of phenotypes [57,58,59]. On one end of the spectrum are classically activated (M1) macrophages that release pro-inflammatory cytokines and oxidative metabolites, thereby facilitating immune function [57,58]. Alternatively, TAMs can be induced to exhibit an immunosuppressive (M2) phenotype [57,59]. Due to various anti-inflammatory factors released by tumor cells, GBM TAMs exist mostly in the M2 state [58,59,60,61]. Upon assuming this state, TAMs can further facilitate tumor development by secreting oncogenic and angiogenic ligands (e.g., TGF-β, EGF, VEGF) as well as ECM-degrading matrix metalloproteinases [59,61]. Moreover, these TAMs can also facilitate immune evasion by inhibiting the anti-tumor activity of other immune cells in the TME [57,58].

Apart from TAMs, tumor-infiltrating lymphocytes (TILs) are also found in relative abundance in the GBM TME [57,58,62]. TILs include CD4^+^ helper T cells, CD8^+^ cytotoxic T lymphocytes (CTLs), and regulatory T cells (T_regs_) [57,61]. Similar to TAMs, TILs mostly exist in an immunosuppressive or inactive state marked by high expression of inhibitory co-receptors (e.g., programmed cell death-1 receptor or PD-1, cytotoxic T lymphocyte-associated protein 4 or CTLA-4) [58,61,62]. These receptors promote T cell inactivation upon binding to their respective ligands, which may be present on the surface of GBM cells or other immune cells like TAMs [57,60,61]. Aside from this mechanism, TILs may also be exhausted by constant exposure to tumor antigens or inhibited by soluble factors secreted by tumor and immune cells [57,58,60,61]. 

Supporting a possible immunoregulatory function of EMP3 in GBM, scRNA-seq of patient samples showed that EMP3 is highly expressed in GBM-infiltrating macrophages and to a lesser extent, T cells [8]. While immunological characterization of EMP3 remains very limited, one study has demonstrated the role of EMP3 in the inhibition of T cell-mediated immunity. Using mouse-derived macrophage cells, Kusumoto et al. have shown that Emp3 overexpression and knockdown reduced and promoted alloreactive induction of CD8+ CTLs, respectively [63]. The inhibition of CTL induction and proliferation by Emp3-overexpressing macrophages was further attributed to increased macrophage TNF-α production, which led to a concomitant downregulation of IL-2Rα on CD8^+^ cells [63] (Figure 9). Moving forward, it will be interesting to explore whether EMP3-high macrophages in the GBM immune microenvironment are able to inhibit tumor-infiltrating CTLs via a similar mechanism. Likewise, it will be relevant to investigate how EMP3 in T cells impacts anti-tumor immunity.

## 5. Summary and Outlook

While the genetic basis of IDH-wt GBM has been comprehensively described, such knowledge has not been fully translated into effective therapies. To combat this bottleneck, there is a need to look beyond obvious genomic alterations and understand other determinants of IDH-wt GBM pathology. In line with this, there has been emerging interest in the role of non-catalytic proteins (e.g., scaffolds, anchors, adaptors, transcription co-factors) in the development of IDH-wt GBM. Although these proteins do not directly perform growth-promoting catalytic functions like deregulated RTKs, their ability to modulate critical nodes of mitogenic signaling pathways still makes them biologically and clinically relevant players in IDH-wt GBM.

In this review, we summarized how the non-catalytic, multifunctional protein EMP3 is able to regulate the activity of several membrane receptors, most of which have been previously associated with IDH-wt GBM. First, EMP3 increases the activity of GBM-relevant RTKs. Furthermore, EMP3 can promote the expression, stability, or activity of several ECM receptors and ECM degraders, thereby outlining how it can contribute to a pro-invasive GBM phenotype. EMP3 also interacts with the purinergic P2RX7 receptor, which is increasingly considered an active contributor to GBM pathology. In addition, EMP3 is emerging as a putative regulator of the immune response elicited by macrophages and T cells.

Taking all this together, EMP3 is an interesting target for therapeutic interference. It could be shown multiple times in different tumor entities that EMP3 knockdown is of therapeutic use in cell culture and animal models [17,25,35,36]. Unfortunately, the molecular and structural basis of how EMP3 exerts its function is unknown. Thus, finding a suitable inhibitor has been impossible until now. However, recent developments in drug discovery have paved for chemical biology approaches that can target “undruggable” proteins like EMP3. One such approach could be the use of PROteolysis TArgeting Chimeras (PROTACs), which are bifunctional molecules that link a protein of interest (POI) to an E3 ubiquitin ligase [64,65]. Using an EMP3-binding PROTAC, EMP3 can potentially be marked for proteasomal degradation with ubiquitin, and the tumor cells’ own proteasome degrades it thereafter. Another alternative would be LYsosome-TArgeting Chimeras (LYTACs) [66,67]. In this very recent technique, the POI is linked by an antibody to a lysosome-shuttling receptor, which fosters internalization and subsequent lysosomal degradation of the targeted protein [67]. This method is especially useful for membrane proteins in combination with antibodies binding the extracellular portion of the protein and thus could be easily used to target EMP3. The only drawback is that these approaches are too novel, so any information on its therapeutic use in humans is still missing. 

In conclusion, further studies will be needed to dissect the contribution and therapeutic potential of EMP3 to IDH-wt GBM. Given the highly versatile nature of this protein and the extreme inter- and intra-tumoral heterogeneity in IDH-wt GBM, it is likely that EMP3 fulfills multiple—and quite possibly subtype-specific—functions in this disease. Comprehensive spatiotemporal proteomics approaches will be necessary to further determine how EMP3 is able to fulfill multiple functions inside the cell. Moreover, gene expression studies using knockdown or knockout models can be utilized to further elucidate oncogenic transcriptional programs that are activated by or are dependent on EMP3. Further immunological explorations will also be necessary to understand how EMP3-expressing malignant cells and immune cells cooperate to promote GBM development in vivo. Lastly, use of targeted protein degradation strategies can be used to model the potential significance of EMP3 as a therapeutic target. Integrating all of these approaches will shed light on how this small membrane protein is able to exert multiple far-reaching oncogenic effects in GBM.

## Figures and Tables

**Figure 1 ijms-22-05261-f001:**
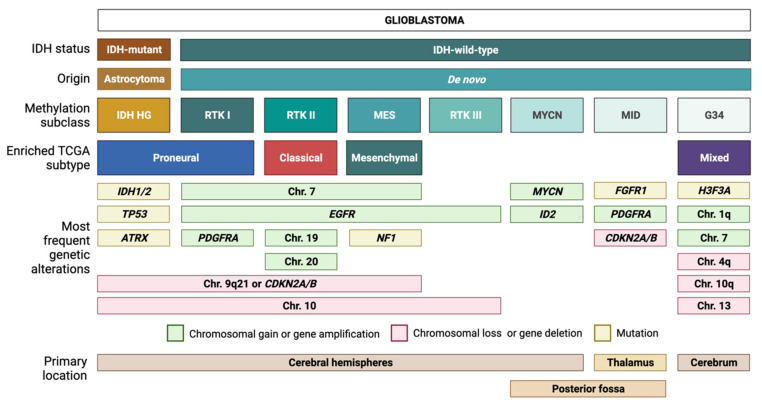
Glioblastoma (GBM) classification. Tumors with a histological diagnosis of GBM can be mainly divided into IDH-mutant GBMs, which typically arise from IDH-mutant astrocytomas, and IDH wild-type GBMs, which usually arise de novo. Based on DNA methylation patterns, IDH-wt GBMs can be further subdivided into several methylation subclasses. Each subclass corresponds to a unique combination of genetic alterations and transcriptomic features.

**Figure 2 ijms-22-05261-f002:**
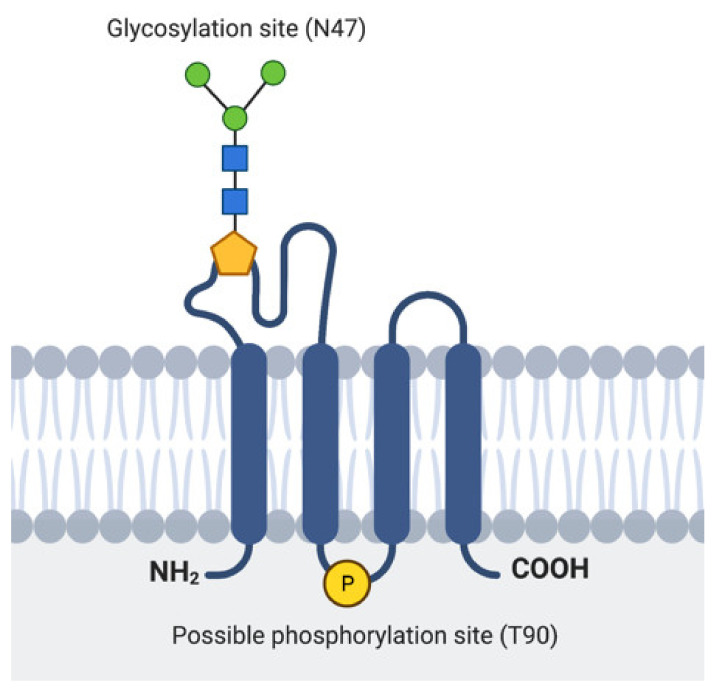
Representation of EMP3′s protein structure. Location of the experimentally verified glycosylation site in the 1st ECD and the predicted phosphorylation site in the intracellular loop are indicated.

**Figure 3 ijms-22-05261-f003:**
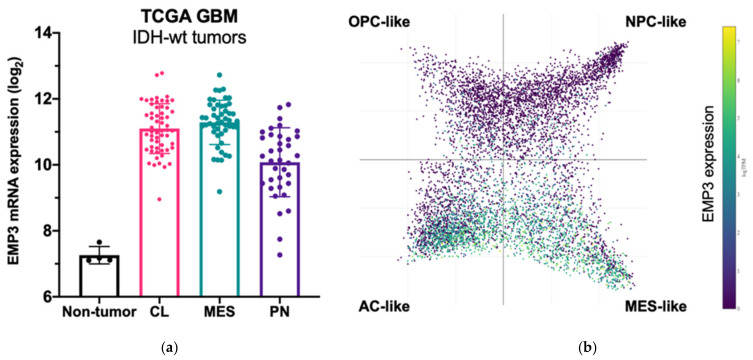
EMP3 expression in glioblastoma (GBM). (**a**) mRNA expression of EMP3 based on RNA sequencing of normal brain tissue and bulk tumor samples. Tumors belong to the TCGA GBM cohort [5] and are stratified according to expression-based subtypes. CL—classical; MES—mesenchymal; PN—proneural; (**b**) single-cell expression pattern of EMP3 in four cellular states based on the publicly available scRNA-seq dataset of GBM tumors [8]. OPC—oligodendrocyte progenitor; NPC—neural progenitor; AC—astrocyte.

**Figure 4 ijms-22-05261-f004:**
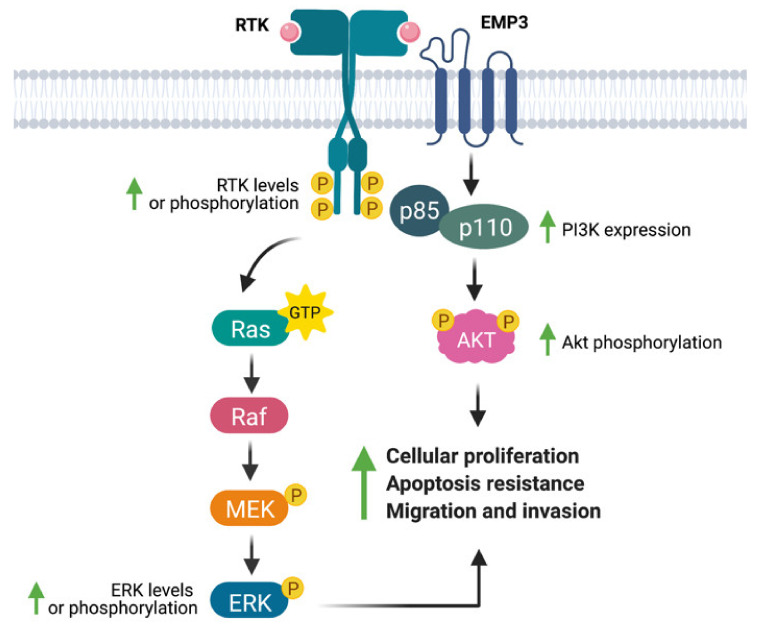
EMP3′s effects on receptor tyrosine kinase (RTK)-dependent mitogenic signaling. EMP3 has been shown to support the expression or phosphorylation of the RTKs EGFR and ErbB2/HER2, as well as their downstream effectors (ERK, PI3K, Akt). These biochemical findings correlate well with increased proliferation, apoptosis resistance, and migration of tumor cells.

**Figure 5 ijms-22-05261-f005:**
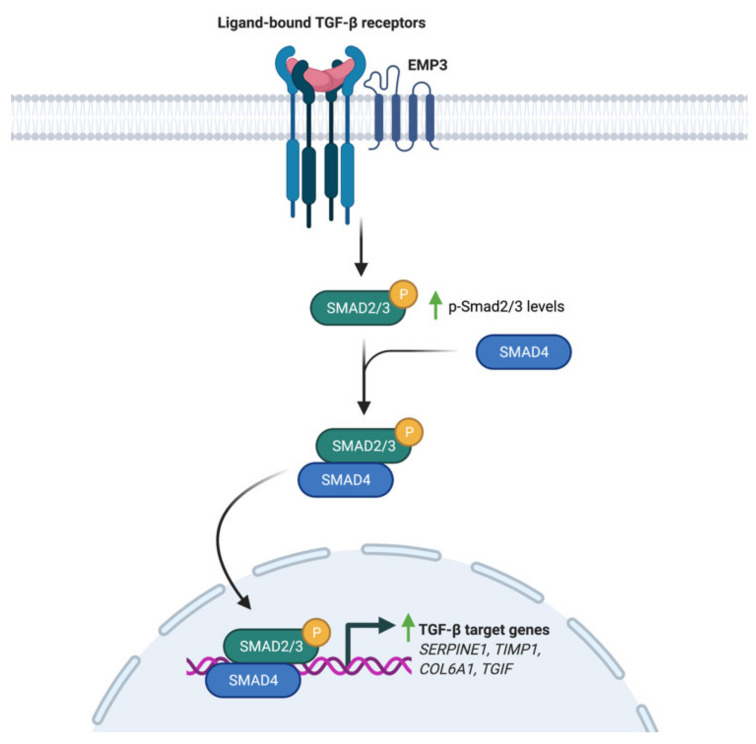
Regulation of TGF-β signaling by EMP3. Upon induction with TGF-β, EMP3 interacts with TGFBR2. This increases Smad 2/3 phosphorylation and transcription of TGF-β target genes.

**Figure 6 ijms-22-05261-f006:**
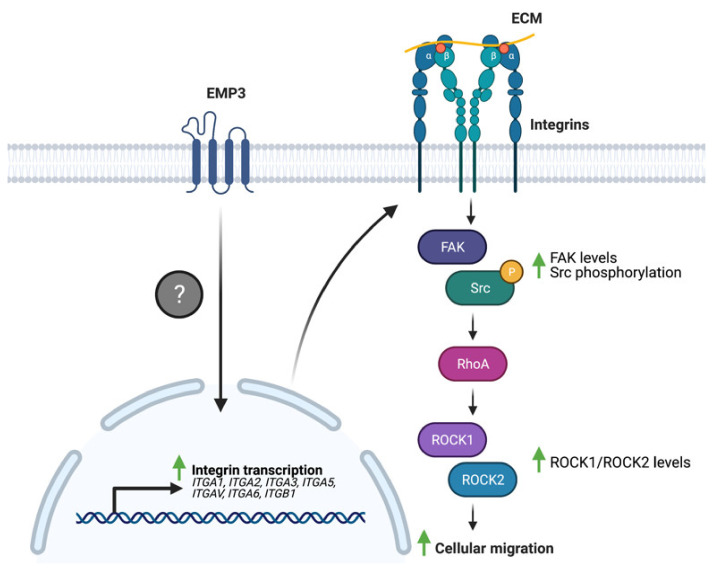
EMP3 and integrin signaling. EMP3 promotes the transcription of several integrins through an unknown mechanism. Enhanced integrin expression, in turn, facilitates cellular migration via FAK-Src signaling.

**Figure 7 ijms-22-05261-f007:**
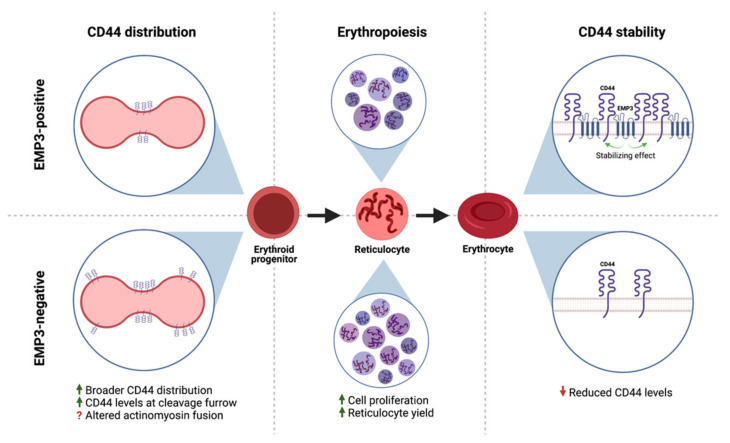
EMP3 and CD44 in erythroid cells. EMP3 affects erythropoiesis by regulating the membrane distribution of CD44 in dividing erythroid progenitors. EMP3 also interacts with and stabilizes CD44 on the membrane of erythrocytes.

**Figure 8 ijms-22-05261-f008:**
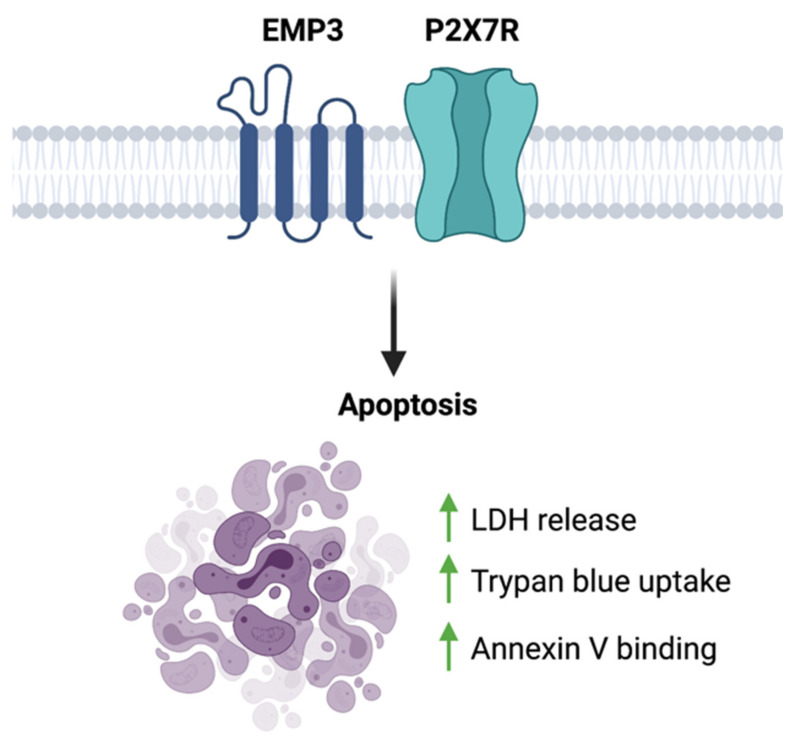
EMP3-P2 × 7R function. EMP3- P2RX7 interaction has been shown to promote apoptosis in HEK293 cells.

**Figure 9 ijms-22-05261-f009:**
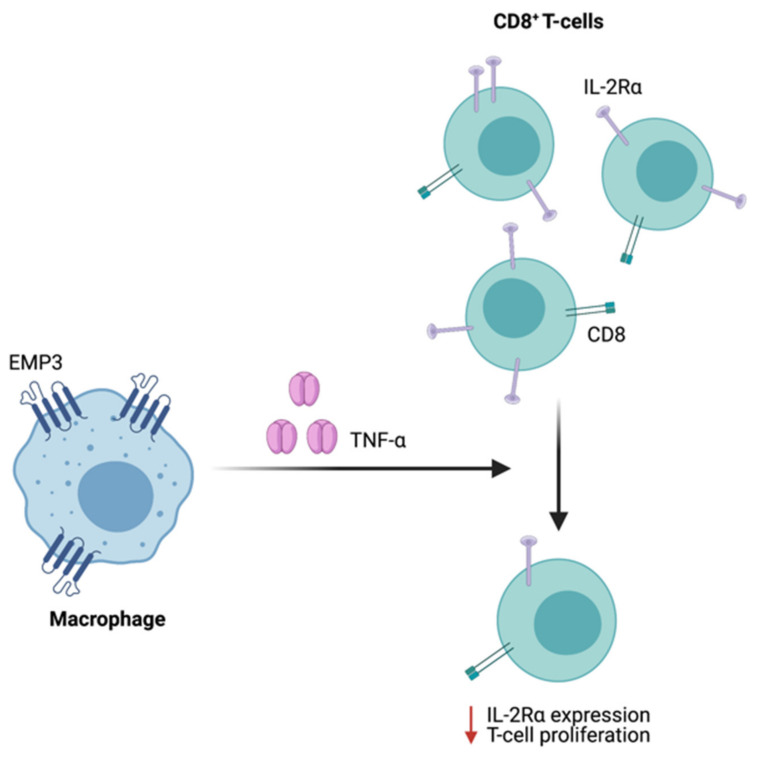
Regulation of CTL function by Emp3. In mice, Emp3-overexpressing macrophages secrete high amounts of TNF-α, leading to IL-2Rα downregulation and reduced CTL induction.

## Data Availability

The data presented in this study are openly available.

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
