# Peer review of "The Multifunctional Role of EMP3 in the Regulation of Membrane Receptors Associated with IDH-Wild-Type Glioblastoma"

_ijms, 2021, doi:10.3390/ijms22105261_

Round 1
Reviewer 1 Report
Revision to the manuscript submitted by Antoni Andreu Martija and Stefan Pusch: The multifunctional role of EMP3 in the regulation of glioblastoma-associated membrane receptors
The authors described EMP3 actions on the IDH-wt glioblastoma (GBM) development and how EMP3 can regulate several membrane proteins that affect the overall survival from patient affected with this GBM subtype. As it was mentioned in the manuscript, EMP3 was established to be part of a 4-gene signature that can predict survival of patients with GBM. Moreover EMP3 was identified to be part of a 60-gene signature that was differentially expressed 159 between long-term (survival > 4 years) and short-term (<1 year) survivors in the Memoral Sloan Kettering Cancer Center and TCGA patient cohorts. Up to date, a review of the role of EMP3 in GBM was missing, therefore I consider this review necessary for new hypothesis for new therapeutic approaches for IDH-wt GBM.
Major comments:
- The review is well structured and easy to follow.
- Since the entire review focused on the IDH-wt GBM, it should be specify on the tittle:
- GBM classification is confusing from the description (line 26-46), will be more intuitive and easy to follow with a schematic description or table.
- Line 50, the term “elsewhere” is not appropriate, please cite the authors.
- Long sentences and grammar need to be revised (i.e. line 182-185)
- Well described interaction between EMP3 and RTK. Are RTK inhibitors a common GBM treatment? If yes, how this affect EMP3 expression and localization in the cell?
- Figure 6 does not increase understanding of CD44 and EMP3 interaction, it is redundant.
- Conclusion presented in line 390-393 is based in pure speculations with no biological background, especially after the statement that EMP3 is presumed to favor the cytolytic nature of P2X7R, EMP3 overexpression (frequently overexpressed in IDH-wt GBM based on several studies) induced membrane blebbing and increased lactate dehydrogenase (LDH) release, trypan blue uptake, and annexin V binding (Citation 56 from the manuscript). Consequently figure 7 is misleading.
- If immune role of EMP3 wants to be addressed, it should be very much extended describing a immune cell specific role of EMP3. Particularly microglia role of EMP3 cells and extracellular vesicles (EVs)-drive communication with tumor cells. If the review then exceed the word limitation, this part should be removed.
- Summary and outlook must be extended and including targeting therapies for EMP3 (i.e. blocking interaction with related proteins)
Minor comments:
- Information regarding glycosylation from line 78-83 (“ Between… immunoblotting.”) is irrelevant since it has not be linked biological/pathological consequence. If the authors consider relevance, please, explain the reason.
- Good selection of references for documentation.
Author Response
Reviewer 1
The authors described EMP3 actions on the IDH-wt glioblastoma (GBM) development and how EMP3 can regulate several membrane proteins that affect the overall survival from patient affected with this GBM subtype. As it was mentioned in the manuscript, EMP3 was established to be part of a 4-gene signature that can predict survival of patients with GBM. Moreover EMP3 was identified to be part of a 60-gene signature that was differentially expressed 159 between long-term (survival > 4 years) and short-term (<1 year) survivors in the Memorial Sloan Kettering Cancer Center and TCGA patient cohorts. Up to date, a review of the role of EMP3 in GBM was missing, therefore I consider this review necessary for new hypothesis for new therapeutic approaches for IDH-wt GBM.
Major comments:
- The review is well structured and easy to follow.
- Thank you for this remark, as it was our primary goal to structure the review in a way which makes it easy to follow and allow a comprehensive overview on a short glance.
- Since the entire review focused on the IDH-wt GBM, it should be specify on the tittle:
- We do agree on this and have changed the title to: “The multifunctional role of EMP3 in the regulation of membrane receptors associated with IDH-wild-type glioblastoma”
- GBM classification is confusing from the description (line 26-46), will be more intuitive and easy to follow with a schematic description or table.
- Diagnostics of brain tumors is very complex due to the manifold of entities. We do agree that a schematic overview of the current diagnostics is helpful at this position and added it as figure 1. We changed a couple of sentences connected to diagnosis to better match this schemata.
- Line 50, the term “elsewhere” is not appropriate, please cite the authors.
- We cited the authors Weller et al. and Pearson et al.
- Long sentences and grammar need to be revised (i.e. line 182-185)
- We changed the sentences to:
The RTK EGFR is frequently overactivated in IDH-wt GBM. EGFR overactivation may result from EGFR gene amplification or the EGFR-vIII mutation, which is defined by an intragenic truncation that leads to constitutive activation of the kinase domain [3,15].
- Well described interaction between EMP3 and RTK. Are RTK inhibitors a common GBM treatment? If yes, how this affect EMP3 expression and localization in the cell?
- We do agree that this is an interesting question, but unfortunately there is no literature on this topic available.
- Figure 6 does not increase understanding of CD44 and EMP3 interaction, it is redundant.
- We changed the figure and the accompanying text to clearly summarize the essential information.
- Conclusion presented in line 390-393 is based in pure speculations with no biological background, especially after the statement that EMP3 is presumed to favor the cytolytic nature of P2X7R, EMP3 overexpression (frequently overexpressed in IDH-wt GBM based on several studies) induced membrane blebbing and increased lactate dehydrogenase (LDH) release, trypan blue uptake, and annexin V binding (Citation 56 from the manuscript). Consequently figure 7 is misleading.
- We rewrote this passage and adjusted the figure to match the cited data more closely. The conclusion now reads:
Whether this pro-apoptotic effect holds true in the setting of GBM remains to be explored. Given the two proteins’ individual links to GBM development, it is attractive to think that in GBM, EMP3 can shift P2X7R to its pro-tumorigenic state instead. Further biochemical and functional studies, however, will be certainly required to test this hypothesis.
- If immune role of EMP3 wants to be addressed, it should be very much extended describing a immune cell specific role of EMP3. Particularly microglia role of EMP3 cells and extracellular vesicles (EVs)-drive communication with tumor cells. If the review then exceed the word limitation, this part should be removed.
- We provided additional background discussing the role of immune cells in the GBM tumor microenvironment. This sets the context as to why the study by Kusumoto et al. mentioned in this section is relevant. However, there are no other studies explaining the role of EMP3 in microglia. We also chose not to discuss EVs as we think this is beyond the scope of the section.
- Summary and outlook must be extended and including targeting therapies for EMP3 (i.e. blocking interaction with related proteins)
- We do agree that targeted therapy options addressing EMP3 are definitely interesting, but so far nothing else than cell culture and animal models have been published. In both model system techniques were used which are unsuitable for human treatment. And as long as the functional domains of EMP3 remain elusive, there is no option of blocking any of EMP3s interactions. Nevertheless we addressed this point by adding some therapeutic interesting approaches in an additional paragraph. These approaches (PROTAC and LYTAC) can be used on EMP3 as they can be utilized without knowledge of functional domains.
Minor comments:
- Information regarding glycosylation from line 78-83 (“ Between… immunoblotting.”) is irrelevant since it has not be linked biological/pathological consequence. If the authors consider relevance, please, explain the reason.
- We do agree that there is no biological function linked to the glycosylation itself so far. But glycosylation plays an essential role in several processes including the immunology. Although these processes may currently not be known for EMP3, we do think it is worth mentioning the glycosylation of EMP3 at this point.
2. Good selection of references for documentation.
Reviewer 2 Report
In this paper Martija and Pusch reviewed the multifunctional role of the tetraspan membrane protein EMP3 in the neurobiology of IDH-wt GMB. In detail, the Authors analyzed the regulatory effects exterted by the non-catalytic, multifunctional protein EMP3 on the activity of several membrane receptors, which are known to be associated with IDH-wt GBM.
Overall, the paper is nicely written, informative, and shed new lights on the possible interpretation regarding the involvement of EMP3 in the aggressive pro-invasive phenotype of IDH-wt GBM. The manuscript is well organized and the cartoons are very useful to render the molecular mechanisms/intracellular pathways clearer to the reader.
Minor point: there are a few places with grammatical errors/spelling errors that are reported below.
Line 21: “The diagnosis OF glioblastoma (GBM)…”
Line 63: “protein 22-kDa (PMP22) gene family BASED ON the sequence..”
Line 81: “the presence of an asparagine at position 47..”
Line 89: “of this predicted PTM remainS to be confirmed.”
Line 119: “human brain, which showS very low EMP3 expression…”
Line 135: “of EMP3 in LGGs correspondS to EMP3 promoter hypermethylation…”
Lines 156-157: “In addition, EMP3 was established to be part of a 4-gene signature that can predict THE survival of patients with GBM [20]…”
Line 160: “long-term (i.e., survival > 4 years) and short-term (i.e., <1 year) survivors..”
Line 163: “TCGA patients with THE survival of less than 1 year. THE mayority…”
Lines 235-236: “its cytoplasmic domains, except for its 13-amino acid intracellular loop
Line 251: “it will be necessary to understand EMP3-depeNdent RTK…”
Lnes 351: “and in a cellular panel consisting OF normal human astrocytes…”
Lines 375-376: “ to increase THE expression of epithelial-to-mesenchymal transition (EMT) markers..”
Line 391: “it is attractive to think that EMP3 can shift P2X7R..”
Line 432: “considered as an active contributor to GBM pathology…”
Author Response
In this paper Martija and Pusch reviewed the multifunctional role of the tetraspan membrane protein EMP3 in the neurobiology of IDH-wt GMB. In detail, the Authors analyzed the regulatory effects exterted by the non-catalytic, multifunctional protein EMP3 on the activity of several membrane receptors, which are known to be associated with IDH-wt GBM.
Overall, the paper is nicely written, informative, and shed new lights on the possible interpretation regarding the involvement of EMP3 in the aggressive pro-invasive phenotype of IDH-wt GBM. The manuscript is well organized and the cartoons are very useful to render the molecular mechanisms/intracellular pathways clearer to the reader.
Minor point: there are a few places with grammatical errors/spelling errors that are reported below.
Line 21: “The diagnosis OF glioblastoma (GBM)…”
Line 63: “protein 22-kDa (PMP22) gene family BASED ON the sequence..”
Line 81: “the presence of an asparagine at position 47..”
Line 89: “of this predicted PTM remainS to be confirmed.”
Line 119: “human brain, which showS very low EMP3 expression…”
Line 135: “of EMP3 in LGGs correspondS to EMP3 promoter hypermethylation…”
Lines 156-157: “In addition, EMP3 was established to be part of a 4-gene signature that can predict THE survival of patients with GBM [20]…”
Line 160: “long-term (i.e., survival > 4 years) and short-term (i.e., <1 year) survivors..”
Line 163: “TCGA patients with THE survival of less than 1 year. THE mayority…”
Lines 235-236: “its cytoplasmic domains, except for its 13-amino acid intracellular loop
Line 251: “it will be necessary to understand EMP3-depeNdent RTK…”
Lnes 351: “and in a cellular panel consisting OF normal human astrocytes…”
Lines 375-376: “ to increase THE expression of epithelial-to-mesenchymal transition (EMT) markers..”
Line 391: “it is attractive to think that EMP3 can shift P2X7R..”
Line 432: “considered as an active contributor to GBM pathology…”
- Thank you for your positive review and proof reading the manuscript so thoroughly. We addressed all your comments accordingly.